# Improved Techniques for Training Score-Based Generative Models

**Yang Song**
Computer Science Department
Stanford University
yangsong@cs.stanford.edu

**Stefano Ermon**
Computer Science Department
Stanford University
ermon@cs.stanford.edu

## Abstract

Score-based generative models can produce high quality image samples comparable to GANs, without requiring adversarial optimization. However, existing training procedures are limited to images of low resolution (typically below $32 \times 32$), and can be unstable under some settings. We provide a new theoretical analysis of learning and sampling from score-based models in high dimensional spaces, explaining existing failure modes and motivating new solutions that generalize across datasets. To enhance stability, we also propose to maintain an exponential moving average of model weights. With these improvements, we can scale score-based generative models to various image datasets, with diverse resolutions ranging from $64 \times 64$ to $256 \times 256$. Our score-based models can generate high-fidelity samples that rival best-in-class GANs on various image datasets, including CelebA, FFHQ, and several LSUN categories.

## 1 Introduction

Score-based generative models [1] represent probability distributions through score—a vector field pointing in the direction where the likelihood of data increases most rapidly. Remarkably, these score functions can be learned from data without requiring adversarial optimization, and can produce realistic image samples that rival GANs on simple datasets such as CIFAR-10 [2].

Despite this success, existing score-based generative models only work on low resolution images ($32 \times 32$) due to several limiting factors. First, the score function is learned via denoising score matching [3, 4, 5]. Intuitively, this means a neural network (named the *score network*) is trained to denoise images blurred with Gaussian noise. A key insight from [1] is to perturb the data using *multiple* noise scales so that the score network captures both coarse and fine-grained image features. However, it is an open question how these noise scales should be chosen. The recommended settings in [1] work well for $32 \times 32$ images, but perform poorly when the resolution gets higher. Second, samples are generated by running Langevin dynamics [6, 7]. This method starts from white noise and progressively denoises it into an image using the score network. This procedure, however, might fail or take an extremely long time to converge when used in high-dimensions and with a necessarily imperfect (learned) score network.

We propose a set of techniques to scale score-based generative models to high resolution images. Based on a new theoretical analysis on a simplified mixture model, we provide a method to analytically compute an effective set of Gaussian noise scales from training data. Additionally, we propose an efficient architecture to amortize the score estimation task across a large (possibly infinite) number of noise scales with a single neural network. Based on a simplified analysis of the convergence properties of the underlying Langevin dynamics sampling procedure, we also derive a technique to approximately optimize its performance as a function of the noise scales. Combining these techniques with an exponential moving average (EMA) of model parameters, we are able to significantly improve

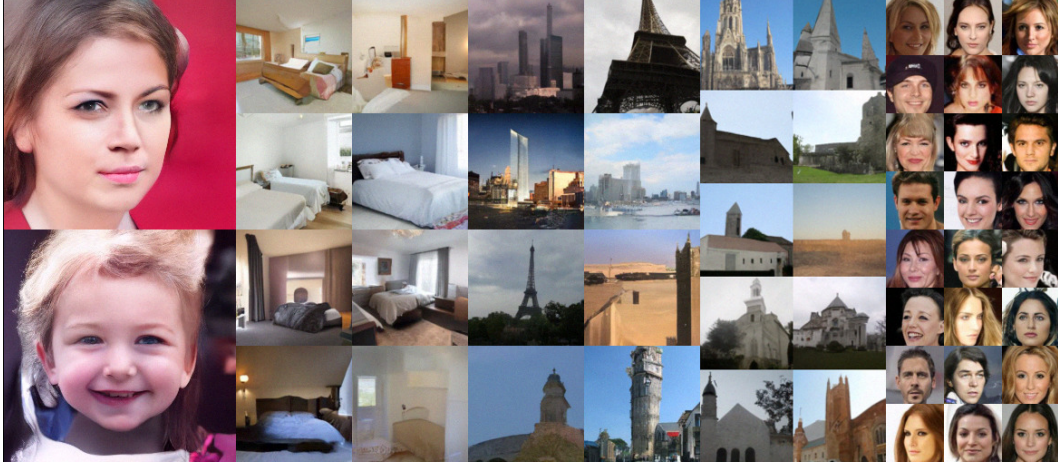

Figure 1: Generated samples on datasets of decreasing resolutions. From left to right: FFHQ $256 \times 256$, LSUN bedroom $128 \times 128$, LSUN tower $128 \times 128$, LSUN church_outdoor $96 \times 96$, and CelebA $64 \times 64$.

the sample quality, and successfully scale to images of resolutions ranging from $64 \times 64$ to $256 \times 256$, which was previously impossible for score-based generative models. As illustrated in Fig. 1, the samples are sharp and diverse.

## 2 Background

### 2.1 Langevin dynamics

For any continuously differentiable probability density $p(\mathbf{x})$, we call $\nabla_{\mathbf{x}} \log p(\mathbf{x})$ its *score function*. In many situations the score function is easier to model and estimate than the original probability density function [3, 8]. For example, for an unnormalized density it does not depend on the partition function. Once the score function is known, we can employ Langevin dynamics to sample from the corresponding distribution. Given a step size $\alpha > 0$, a total number of iterations $T$, and an initial sample $\mathbf{x}_0$ from any prior distribution $\pi(\mathbf{x})$, Langevin dynamics iteratively evaluate the following

$$\mathbf{x}_t \leftarrow \mathbf{x}_{t-1} + \alpha \, \nabla_{\mathbf{x}} \log p(\mathbf{x}_{t-1}) + \sqrt{2\alpha} \, \mathbf{z}_t, \quad 1 \leq t \leq T \tag{1}$$

where $\mathbf{z}_t \sim \mathcal{N}(\mathbf{0}, \mathbf{I})$. When $\alpha$ is sufficiently small and $T$ is sufficiently large, the distribution of $\mathbf{x}_T$ will be close to $p(\mathbf{x})$ under some regularity conditions [6, 7]. Suppose we have a neural network $\mathbf{s}_{\boldsymbol{\theta}}(\mathbf{x})$ (called the *score network*) parameterized by $\boldsymbol{\theta}$, and it has been trained such that $\mathbf{s}_{\boldsymbol{\theta}}(\mathbf{x}) \approx \nabla_{\mathbf{x}} \log p(\mathbf{x})$. We can approximately generate samples from $p(\mathbf{x})$ using Langevin dynamics by replacing $\nabla_{\mathbf{x}} \log p(\mathbf{x}_{t-1})$ with $\mathbf{s}_{\boldsymbol{\theta}}(\mathbf{x}_{t-1})$ in Eq. (1). Note that Eq. (1) can be interpreted as noisy gradient ascent on the log-density $\log p(\mathbf{x})$.

### 2.2 Score-based generative modeling

We can estimate the score function from data and generate new samples with Langevin dynamics. This idea was named *score-based generative modeling* by ref. [1]. Because the estimated score function is inaccurate in regions without training data, Langevin dynamics may not converge correctly when a sampling trajectory encounters those regions (see more detailed analysis in ref. [1]). As a remedy, ref. [1] proposes to perturb the data with Gaussian noise of different intensities and jointly estimate the score functions of all noise-perturbed data distributions. During inference, they combine the information from all noise scales by sampling from each noise-perturbed distribution sequentially with Langevin dynamics.

More specifically, suppose we have an underlying data distribution $p_{\text{data}}(\mathbf{x})$ and consider a sequence of noise scales $\{\sigma_i\}_{i=1}^{L}$ that satisfies $\sigma_1 > \sigma_2 > \cdots > \sigma_L$. Let $p_\sigma(\tilde{\mathbf{x}} \mid \mathbf{x}) = \mathcal{N}(\tilde{\mathbf{x}} \mid \mathbf{x}, \sigma^2 \mathbf{I})$, and denote the corresponding perturbed data distribution as $p_\sigma(\tilde{\mathbf{x}}) \triangleq \int p_\sigma(\tilde{\mathbf{x}} \mid \mathbf{x}) p_{\text{data}}(\mathbf{x}) \mathrm{d}\mathbf{x}$. Ref. [1]

proposes to estimate the score function of each $p_{\sigma_i}(\mathbf{x})$ by training a joint neural network $\mathbf{s}_{\boldsymbol{\theta}}(\mathbf{x}, \sigma)$ (called the *noise conditional score network*) with the following loss:

$$\frac{1}{2L} \sum_{i=1}^{L} \mathbb{E}_{p_{\text{data}}(\mathbf{x})} \mathbb{E}_{p_{\sigma_i}(\tilde{\mathbf{x}}|\mathbf{x})} \left[ \left\| \sigma_i \mathbf{s}_{\boldsymbol{\theta}}(\tilde{\mathbf{x}}, \sigma_i) + \frac{\tilde{\mathbf{x}} - \mathbf{x}}{\sigma_i} \right\|_2^2 \right], \tag{2}$$

where all expectations can be efficiently estimated using empirical averages. When trained to the optimum (denoted as $s_{\boldsymbol{\theta}^*}(\mathbf{x}, \sigma)$), the noise conditional score network (NCSN) satisfies $\forall i$ : $s_{\boldsymbol{\theta}^*}(\mathbf{x}, \sigma_i) = \nabla_{\mathbf{x}} \log p_{\sigma_i}(\mathbf{x})$ almost everywhere [1], assuming enough data and model capacity.

After training an NCSN, ref. [1] generates samples by *annealed Langevin dynamics*, a method that combines information from all noise scales. We provide its pseudo-code in Algorithm 1. The approach amounts to sampling from $p_{\sigma_1}(\mathbf{x}), p_{\sigma_2}(\mathbf{x}), \cdots, p_{\sigma_L}(\mathbf{x})$ sequentially with Langevin dynamics with a special step size schedule $\alpha_i = \epsilon \, \sigma_i^2/\sigma_L^2$ for the $i$-th noise scale. Samples from each noise scale are used to initialize Langevin dynamics for the next noise scale until reaching the smallest one, where it provides final samples for the NCSN.

---

**Algorithm 1** Annealed Langevin dynamics [1]

---

**Require:** $\{\sigma_i\}_{i=1}^{L}, \epsilon, T$.
1: Initialize $\mathbf{x}_0$
2: **for** $i \leftarrow 1$ to $L$ **do**
3:     $\alpha_i \leftarrow \epsilon \cdot \sigma_i^2/\sigma_L^2$        $\triangleright \alpha_i$ is the step size.
4:     **for** $t \leftarrow 1$ to $T$ **do**
5:         Draw $\mathbf{z}_t \sim \mathcal{N}(0, I)$
6:         $\mathbf{x}_t \leftarrow \mathbf{x}_{t-1} + \alpha_i \, \mathbf{s}_{\boldsymbol{\theta}}(\mathbf{x}_{t-1}, \sigma_i) + \sqrt{2\alpha_i} \, \mathbf{z}_t$
7:     $\mathbf{x}_0 \leftarrow \mathbf{x}_T$
8: **if** denoise $\mathbf{x}_T$ **then**
9:     **return** $\mathbf{x}_T + \sigma_T^2 \mathbf{s}_{\boldsymbol{\theta}}(\mathbf{x}_T, \sigma_T)$
10: **else**
11:     **return** $\mathbf{x}_T$

---

Following the first public release of this work, ref. [9] noticed that adding an extra denoising step after the original annealed Langevin dynamics in [1], similar to [10, 11, 12], often significantly improves FID scores [13] without affecting the visual appearance of samples. Instead of directly returning $\mathbf{x}_T$, this denoising step returns $\mathbf{x}_T + \sigma_T^2 \mathbf{s}_{\boldsymbol{\theta}}(\mathbf{x}_T, \sigma_T)$ (see Algorithm 1), which essentially removes the unwanted noise $\mathcal{N}(\mathbf{0}, \sigma_T^2 \mathbf{I})$ from $\mathbf{x}_T$ using Tweedie's formula [14]. Therefore, we have updated results in the main paper by incorporating this denoising trick, but kept some original results without this denoising step in the appendix for reference.

There are many design choices that are critical to the successful training and inference of NCSNs, including (i) the set of noise scales $\{\sigma_i\}_{i=1}^{L}$, (ii) the way that $\mathbf{s}_{\boldsymbol{\theta}}(\mathbf{x}, \sigma)$ incorporates information of $\sigma$, (iii) the step size parameter $\epsilon$ and (iv) the number of sampling steps per noise scale $T$ in Algorithm 1. Below we provide theoretically motivated ways to configure them without manual tuning, which significantly improve the performance of NCSNs on high resolution images.

## 3 Choosing noise scales

Noise scales are critical for the success of NCSNs. As shown in [1], score networks trained with a single noise can never produce convincing samples for large images. Intuitively, high noise facilitates the estimation of score functions, but also leads to corrupted samples; while lower noise gives clean samples but makes score functions harder to estimate. One should therefore leverage different noise scales together to get the best of both worlds.

When the range of pixel values is $[0, 1]$, the original work on NCSN [1] recommends choosing $\{\sigma_i\}_{i=1}^{L}$ as a geometric sequence where $L = 10$, $\sigma_1 = 1$, and $\sigma_L = 0.01$. It is reasonable that the smallest noise scale $\sigma_L = 0.01 \ll 1$, because we sample from perturbed distributions with descending noise scales and we want to add low noise at the end. However, some important questions remain unanswered, which turn out to be critical to the success of NCSNs on high resolution images: (i) Is $\sigma_1 = 1$ appropriate? If not, how should we adjust $\sigma_1$ for different datasets? (ii) Is geometric progression a good choice? (iii) Is $L = 10$ good across different datasets? If not, how many noise scales are ideal?

Below we provide answers to the above questions, motivated by theoretical analyses on simple mathematical models. Our insights are effective for configuring score-based generative modeling in practice, as corroborated by experimental results in Section 6.

## 3.1 Initial noise scale

The algorithm of annealed Langevin dynamics (Algorithm 1) is an iterative refining procedure that starts from generating coarse samples with rich variation under large noise, before converging to fine samples with less variation under small noise. The initial noise scale $\sigma_1$ largely controls the diversity of the final samples. In order to promote sample diversity, we might want to choose $\sigma_1$ to be as large as possible. However, an excessively large $\sigma_1$ will require more noise scales (to be discussed in Section 3.2) and make annealed Langevin dynamics more expensive. Below we present an analysis to guide the choice of $\sigma_1$ and provide a technique to strike the right balance.

Real-world data distributions are complex and hard to analyze, so we approximate them with empirical distributions. Suppose we have a dataset $\{\mathbf{x}^{(1)}, \mathbf{x}^{(2)}, \cdots, \mathbf{x}^{(N)}\}$ which is i.i.d. sampled from $p_{\text{data}}(\mathbf{x})$. Assuming $N$ is sufficiently large, we have $p_{\text{data}}(\mathbf{x}) \approx \hat{p}_{\text{data}}(\mathbf{x}) \triangleq \frac{1}{N} \sum_{i=1}^{N} \delta(\mathbf{x} = \mathbf{x}^{(i)})$, where $\delta(\cdot)$ denotes a point mass distribution. When perturbed with $\mathcal{N}(\mathbf{0}, \sigma_1^2 \mathbf{I})$, the empirical distribution becomes $\hat{p}_{\sigma_1}(\mathbf{x}) \triangleq \frac{1}{N} \sum_{i=1}^{N} p^{(i)}(\mathbf{x})$, where $p^{(i)}(\mathbf{x}) \triangleq \mathcal{N}(\mathbf{x} \mid \mathbf{x}^{(i)}, \sigma_1^2 \mathbf{I})$. For generating diverse samples regardless of initialization, we naturally expect that Langevin dynamics can explore any component $p^{(i)}(\mathbf{x})$ when initialized from any other component $p^{(j)}(\mathbf{x})$, where $i \neq j$. The performance of Langevin dynamics is governed by the score function $\nabla_{\mathbf{x}} \log \hat{p}_{\sigma_1}(\mathbf{x})$ (see Eq. (1)).

**Proposition 1.** *Let $\hat{p}_{\sigma_1}(\mathbf{x}) \triangleq \frac{1}{N} \sum_{i=1}^{N} p^{(i)}(\mathbf{x})$, where $p^{(i)}(\mathbf{x}) \triangleq \mathcal{N}(\mathbf{x} \mid \mathbf{x}^{(i)}, \sigma_1^2 \mathbf{I})$. With $r^{(i)}(\mathbf{x}) \triangleq \frac{p^{(i)}(\mathbf{x})}{\sum_{k=1}^{N} p^{(k)}(\mathbf{x})}$, the score function is $\nabla_{\mathbf{x}} \log \hat{p}_{\sigma_1}(\mathbf{x}) = \sum_{i=1}^{N} r^{(i)}(\mathbf{x}) \nabla_{\mathbf{x}} \log p^{(i)}(\mathbf{x})$. Moreover,*

$$\mathbb{E}_{p^{(i)}(\mathbf{x})}[r^{(j)}(\mathbf{x})] \leq \frac{1}{2} \exp\left( - \frac{\left\| \mathbf{x}^{(i)} - \mathbf{x}^{(j)} \right\|_2^2}{8\sigma_1^2} \right). \tag{3}$$

In order for Langevin dynamics to transition from $p^{(i)}(\mathbf{x})$ to $p^{(j)}(\mathbf{x})$ easily for $i \neq j$, $\mathbb{E}_{p^{(i)}(\mathbf{x})}[r^{(j)}(\mathbf{x})]$ has to be relatively large, because otherwise $\nabla_{\mathbf{x}} \log \hat{p}_{\sigma_1}(\mathbf{x}) = \sum_{k=1}^{N} r^{(k)}(\mathbf{x}) \nabla_{\mathbf{x}} \log p^{(k)}(\mathbf{x})$ will ignore the component $p^{(j)}(\mathbf{x})$ (on average) when initialized with $\mathbf{x} \sim p^{(i)}(\mathbf{x})$ and in such case Langevin dynamics will act as if $p^{(j)}(\mathbf{x})$ did not exist. The bound of Eq. (3) indicates that $\mathbb{E}_{p^{(i)}(\mathbf{x})}[r^{(j)}(\mathbf{x})]$ can decay exponentially fast if $\sigma_1$ is small compared to $\left\| \mathbf{x}^{(i)} - \mathbf{x}^{(j)} \right\|_2$. As a result, it is necessary for $\sigma_1$ to be numerically comparable to the maximum pairwise distances of data to facilitate transitioning of Langevin dynamics and hence improving sample diversity. In particular, we suggest:

**Technique 1** (Initial noise scale). *Choose $\sigma_1$ to be as large as the maximum Euclidean distance between all pairs of training data points.*

Taking CIFAR-10 as an example, the median pairwise distance between all training images is around 18, so $\sigma_1 = 1$ as in [1] implies $\mathbb{E}[r(\mathbf{x})] < 10^{-17}$ and is unlikely to produce diverse samples as per our analysis. To test whether choosing $\sigma_1$ according to Technique 1 (*i.e.*, $\sigma_1 = 50$) gives significantly more diverse samples than using $\sigma_1 = 1$, we run annealed Langevin dynamics to sample from a mixture of Gaussian with 10000 components, where each component is centered at one CIFAR-10 test image. All initial samples are drawn from a uni-

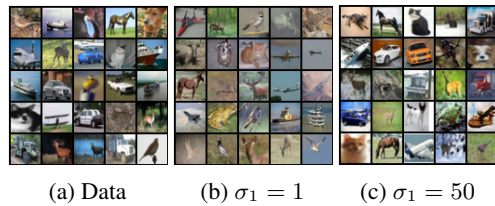

|       (a) Data       |   (b) $\sigma_1 = 1$   |   (c) $\sigma_1 = 50$   |

Figure 2: Running annealed Langevin dynamics to sample from a mixture of Gaussian centered at images in the CIFAR-10 test set.

form distribution over $[0, 1]^{32 \times 32 \times 3}$. This setting allows us to avoid confounders introduced by NCSN training because we use ground truth score functions. As shown in Fig. 2, samples in Fig. 2c (using Technique 1) exhibit comparable diversity to ground-truth images (Fig. 2a), and have better variety than Fig. 2b ($\sigma_1 = 1$). Quantitatively, the average pairwise distance of samples in Fig. 2c is 18.65, comparable to data (17.78) but much higher than that of Fig. 2b (10.12).

## 3.2 Other noise scales

After setting $\sigma_L$ and $\sigma_1$, we need to choose the number of noise scales $L$ and specify the other elements of $\{\sigma_i\}_{i=1}^{L}$. As analyzed in [1], it is crucial for the success of score-based generative models

to ensure that $p_{\sigma_i}(\mathbf{x})$ generates a sufficient number of training data in high density regions of $p_{\sigma_{i-1}}(\mathbf{x})$ for all $1 < i \le L$. The intuition is we need reliable gradient signals for $p_{\sigma_i}(\mathbf{x})$ when initializing Langevin dynamics with samples from $p_{\sigma_{i-1}}(\mathbf{x})$.

However, an extensive grid search on $\{\sigma_i\}_{i=1}^{L}$ can be very expensive. To give some theoretical guidance on finding good noise scales, we consider a simple case where the dataset contains only one data point, or equivalently, $\forall 1 \le i \le L : p_{\sigma_i}(\mathbf{x}) = \mathcal{N}(\mathbf{x} \mid \mathbf{0}, \sigma_i^2 \mathbf{I})$. Our first step is to understand the distributions of $p_{\sigma_i}(\mathbf{x})$ better, especially when $\mathbf{x}$ has high dimensionality. We can decompose $p_{\sigma_i}(\mathbf{x})$ in hyperspherical coordinates to $p(\phi)p_{\sigma_i}(r)$, where $r$ and $\phi$ denote the radial and angular coordinates of $\mathbf{x}$ respectively. Because $p_{\sigma_i}(\mathbf{x})$ is an isotropic Gaussian, the angular component $p(\phi)$ is uniform and shared across all noise scales. As for $p_{\sigma_i}(r)$, we have the following

**Proposition 2.** *Let $\mathbf{x} \in \mathbb{R}^D \sim \mathcal{N}(\mathbf{0}, \sigma^2 \mathbf{I})$, and $r = \|\mathbf{x}\|_2$. We have*

$$p(r) = \frac{1}{2^{D/2-1}\Gamma(D/2)} \frac{r^{D-1}}{\sigma^D} \exp\left(-\frac{r^2}{2\sigma^2}\right) \quad and \quad r - \sqrt{D}\sigma \xrightarrow{d} \mathcal{N}(0, \sigma^2/2) \ when \ D \to \infty.$$

In practice, dimensions of image data can range from several thousand to millions, and are typically large enough to warrant $p(r) \approx \mathcal{N}(r|\sqrt{D}\sigma, \sigma^2/2)$ with negligible error. We therefore take $p_{\sigma_i}(r) = \mathcal{N}(r|m_i, s_i^2)$ to simplify our analysis, where $m_i \triangleq \sqrt{D}\sigma$, and $s_i^2 \triangleq \sigma^2/2$.

Recall that our goal is to make sure samples from $p_{\sigma_i}(\mathbf{x})$ will cover high density regions of $p_{\sigma_{i-1}}(\mathbf{x})$. Because $p(\phi)$ is shared across all noise scales, $p_{\sigma_i}(\mathbf{x})$ already covers the angular component of $p_{\sigma_{i-1}}(\mathbf{x})$. Therefore, we need the radial components of $p_{\sigma_i}(\mathbf{x})$ and $p_{\sigma_{i-1}}(\mathbf{x})$ to have large overlap. Since $p_{\sigma_{i-1}}(r)$ has high density in $\mathcal{I}_{i-1} \triangleq [m_{i-1} - 3s_{i-1}, m_{i-1} + 3s_{i-1}]$ (employing the "three-sigma rule of thumb" [15]), a natural choice is to fix $p_{\sigma_i}(r \in \mathcal{I}_{i-1}) = \Phi(\sqrt{2D}(\gamma_i - 1) + 3\gamma_i) - \Phi(\sqrt{2D}(\gamma_i - 1) - 3\gamma_i) = C$ with some moderately large constant $C > 0$ for all $1 < i \le L$, where $\gamma_i \triangleq \sigma_{i-1}/\sigma_i$ and $\Phi(\cdot)$ is the CDF of standard Gaussian. This choice immediately implies that $\gamma_2 = \gamma_3 = \cdots \gamma_L$ and thus $\{\sigma_i\}_{i=1}^{L}$ is a geometric progression.

Ideally, we should choose as many noise scales as possible to make $C \approx 1$. However, having too many noise scales will make sampling very costly, as we need to run Langevin dynamics for each noise scale in sequence. On the other hand, $L = 10$ (for $32 \times 32$ images) as in the original setting of [1] is arguably too small, for which $C = 0$ up to numerical precision. To strike a balance, we recommend $C \approx 0.5$ which performs well in our experiments. In summary,

**Technique 2** (Other noise scales)**.** *Choose $\{\sigma_i\}_{i=1}^{L}$ as a geometric progression with common ratio $\gamma$, such that $\Phi(\sqrt{2D}(\gamma - 1) + 3\gamma) - \Phi(\sqrt{2D}(\gamma - 1) - 3\gamma) \approx 0.5$.*

### 3.3  Incorporating the noise information

For high resolution images, we need a large $\sigma_1$ and a huge number of noise scales as per Technique 1 and 2. Recall that the NCSN is a single amortized network that takes a noise scale and gives the corresponding score. In [1], authors use a separate set of scale and bias parameters in normalization layers to incorporate the information from each noise scale. However, its memory consumption grows linearly w.r.t. $L$, and it is not applicable when the NCSN has no normalization layers.

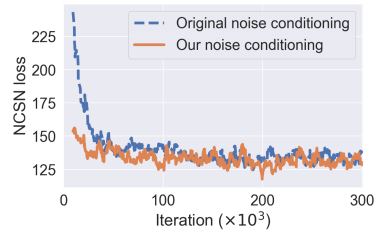

Figure 3: Training loss curves of two noise conditioning methods.

We propose an efficient alternative that is easier to implement and more widely applicable. For $p_\sigma(\mathbf{x}) = \mathcal{N}(\mathbf{x} \mid \mathbf{0}, \sigma^2 \mathbf{I})$ analyzed in Section 3.2, we observe that $\mathbb{E}[\|\nabla_{\mathbf{x}} \log p_\sigma(\mathbf{x})\|_2] \approx \sqrt{D}/\sigma$. Moreover, as empirically noted in [1], $\|\mathbf{s}_{\boldsymbol{\theta}}(\mathbf{x}, \sigma)\|_2 \propto 1/\sigma$ for a trained NCSN on real data. Because the norm of score functions scales inverse proportionally to $\sigma$, we can incorporate the noise information by rescaling the output of an unconditional score network $\mathbf{s}_{\boldsymbol{\theta}}(\mathbf{x})$ with $1/\sigma$. This motivates our following recommendation

**Technique 3** (Noise conditioning)**.** *Parameterize the NCSN with $\mathbf{s}_{\boldsymbol{\theta}}(\mathbf{x}, \sigma) = \mathbf{s}_{\boldsymbol{\theta}}(\mathbf{x})/\sigma$, where $\mathbf{s}_{\boldsymbol{\theta}}(\mathbf{x})$ is an unconditional score network.*

It is typically hard for deep networks to automatically learn this rescaling, because $\sigma_1$ and $\sigma_L$ can differ by several orders of magnitude. This simple choice is easier to implement, and can easily

handle a large number of noise scales (even continuous ones). As shown in Fig. 3 (detailed settings in Appendix B), it achieves similar training losses compared to the original noise conditioning approach in [1], and generate samples of better quality (see Appendix C.4).

## 4 Configuring annealed Langevin dynamics

In order to sample from an NCSN with annealed Langevin dynamics, we need to specify the number of sampling steps per noise scale $T$ and the step size parameter $\epsilon$ in Algorithm 1. Authors of [1] recommends $\epsilon = 2 \times 10^{-5}$ and $T = 100$. It remains unclear how we should change $\epsilon$ and $T$ for different sets of noise scales.

To gain some theoretical insight, we revisit the setting in Section 3.2 where the dataset has one point (*i.e.*, $p_{\sigma_i}(\mathbf{x}) = \mathcal{N}(\mathbf{x} \mid \mathbf{0}, \sigma_i^2 \mathbf{I})$). Annealed Langevin dynamics connect two adjacent noise scales $\sigma_{i-1} > \sigma_i$ by initializing the Langevin dynamics for $p_{\sigma_i}(\mathbf{x})$ with samples obtained from $p_{\sigma_{i-1}}(\mathbf{x})$. When applying Langevin dynamics to $p_{\sigma_i}(\mathbf{x})$, we have $\mathbf{x}_{t+1} \leftarrow \mathbf{x}_t + \alpha \nabla_{\mathbf{x}} \log p_{\sigma_i}(\mathbf{x}_t) + \sqrt{2\alpha}\mathbf{z}_t$, where $\mathbf{x}_0 \sim p_{\sigma_{i-1}}(\mathbf{x})$ and $\mathbf{z}_t \sim \mathcal{N}(\mathbf{0}, \mathbf{I})$. The distribution of $\mathbf{x}_T$ can be computed in closed form:

**Proposition 3.** *Let* $\gamma = \frac{\sigma_{i-1}}{\sigma_i}$. *For* $\alpha = \epsilon \cdot \frac{\sigma_i^2}{\sigma_L^2}$ *(as in Algorithm 1), we have* $\mathbf{x}_T \sim \mathcal{N}(\mathbf{0}, s_T^2 \mathbf{I})$, *where*

$$\frac{s_T^2}{\sigma_i^2} = \left(1 - \frac{\epsilon}{\sigma_L^2}\right)^{2T} \left(\gamma^2 - \frac{2\epsilon}{\sigma_L^2 - \sigma_L^2 \left(1 - \frac{\epsilon}{\sigma_L^2}\right)^2}\right) + \frac{2\epsilon}{\sigma_L^2 - \sigma_L^2 \left(1 - \frac{\epsilon}{\sigma_L^2}\right)^2}. \tag{4}$$

When $\{\sigma_i\}_{i=1}^L$ is a geometric progression as advocated by Technique 2, we immediately see that $s_T^2/\sigma_i^2$ is identical across all $1 < i \leq T$ because of the shared $\gamma$. Furthermore, the value of $s_T^2/\sigma_i^2$ has no explicit dependency on the dimensionality $D$.

For better mixing of annealed Langevin dynamics, we hope $s_T^2/\sigma_i^2$ approaches 1 across all noise scales, which can be achieved by finding $\epsilon$ and $T$ that minimize the difference between Eq. (4) and 1. Unfortunately, this often results in an unnecessarily large $T$ that makes sampling very expensive for large $L$. As an alternative, we propose to first choose $T$ based on a reasonable computing budget (typically $T \times L$ is several thousand), and subsequently find $\epsilon$ by making Eq. (4) as close to 1 as possible. In summary:

**Technique 4** (selecting $T$ and $\epsilon$). *Choose $T$ as large as allowed by a computing budget and then select an $\epsilon$ that makes Eq.* (4) *maximally close to 1.*

We follow this guidance to generate all samples in this paper, except for those from the original NCSN where we adopt the same settings as in [1]. When finding $\epsilon$ with Technique 4 and Eq. (4), we recommend performing grid search over $\epsilon$, rather than using gradient-based optimization methods.

## 5 Improving stability with moving average

Unlike GANs, score-based generative models have one unified objective (Eq. (2)) and require no adversarial training. However, even though the loss function of NCSNs typically decreases steadily over the course of training, we observe that the generated image samples sometimes exhibit unstable visual quality, especially for images of larger resolutions. We empirically demonstrate this fact by training NCSNs on CIFAR-10 $32 \times 32$ and CelebA [16] $64 \times 64$ following the settings of [1], which exemplifies typical behavior on other image datasets. We report FID scores [13] computed on 1000 samples every 5000 iterations. Results in Fig. 4 are computed with the denoising step, but results without the denoising step are similar (see Fig. 8 in Appendix C.1). As shown in Figs. 4 and 8, the FID scores for the vanilla NCSN often fluctuate significantly during training. Additionally, samples from the vanilla NCSN sometimes exhibit characteristic artifacts: image samples from the same checkpoint have strong tendency to have a common color shift. Moreover, samples are shifted towards different colors throughout training. We provide more samples in Appendix C.3 to manifest this artifact.

This issue can be easily fixed by exponential moving average (EMA). Specifically, let $\boldsymbol{\theta}_i$ denote the parameters of an NCSN after the $i$-th training iteration, and $\boldsymbol{\theta}'$ be an independent copy of the parameters. We update $\boldsymbol{\theta}'$ with $\boldsymbol{\theta}' \leftarrow m\boldsymbol{\theta}' + (1 - m)\boldsymbol{\theta}_i$ after each optimization step, where $m$ is the

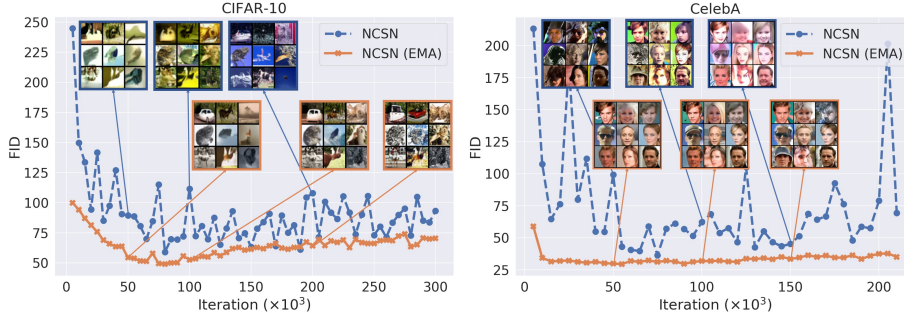

Figure 4: FIDs and color artifacts over the course of training (best viewed in color). The FIDs of NCSN have much higher volatility compared to NCSN with EMA. Samples from the vanilla NCSN often have obvious color shifts. All FIDs are computed with the denoising step.

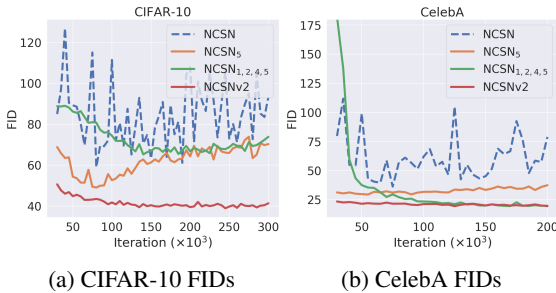

(a) CIFAR-10 FIDs      (b) CelebA FIDs

Figure 5: FIDs for different groups of techniques. Subscripts of "NCSN" are IDs of techniques in effect. "NCSNv2" uses all techniques. Results are computed with the denoising step.

Table 1: Inception and FID scores.

| Model | Inception $\uparrow$ | FID $\downarrow$ |
|---|---|---|
| **CIFAR-10 Unconditional** | | |
| PixelCNN [17] | 4.60 | 65.93 |
| IGEBM [18] | 6.02 | 40.58 |
| WGAN-GP [19] | $7.86 \pm .07$ | 36.4 |
| SNGAN [20] | $8.22 \pm .05$ | 21.7 |
| NCSN [1] | $\mathbf{8.87 \pm .12}$ | 25.32 |
| NCSN (w/ denoising) | $7.32 \pm .12$ | 29.8 |
| NCSNv2 (w/o denoising) | $8.73 \pm .13$ | 31.75 |
| NCSNv2 (w/ denoising) | $8.40 \pm .07$ | **10.87** |
| **CelebA $64 \times 64$** | | |
| NCSN (w/o denoising) | - | 26.89 |
| NCSN (w/ denoising) | - | 25.30 |
| NCSNv2 (w/o denoising) | - | 28.86 |
| NCSNv2 (w/ denoising) | - | **10.23** |

momentum parameter and typically $m = 0.999$. When producing samples, we use $\mathbf{s}_{\boldsymbol{\theta}'}(\mathbf{x}, \sigma)$ instead of $\mathbf{s}_{\boldsymbol{\theta}_i}(\mathbf{x}, \sigma)$. As shown in Fig. 4, EMA can effectively stabilize FIDs, remove artifacts (more samples in Appendix C.3) and give better FID scores in most cases. Empirically, we observe the effectiveness of EMA is universal across a large number of different image datasets. As a result, we recommend the following rule of thumb:

**Technique 5** (EMA). *Apply exponential moving average to parameters when sampling.*

## 6 Combining all techniques together

Employing Technique 1–5, we build NCSNs that can readily work across a large number of different datasets, including high resolution images that were previously out of reach with score-based generative modeling. Our modified model is named NCSNv2. For a complete description on experimental details and more results, please refer to Appendix B and C.

**Quantitative results:** We consider CIFAR-10 $32 \times 32$ and CelebA $64 \times 64$ where NCSN and NCSNv2 both produce reasonable samples. We report FIDs (lower is better) every 5000 iterations of training on 1000 samples and give results in Fig. 5 (with denoising) and Fig. 9 (without denoising, deferred to Appendix C.1). As shown in Figs. 5 and 9, we observe that the FID scores of NCSNv2 (with all techniques applied) are on average better than those of NCSN, and have much smaller variance over the course of training. Following [1], we select checkpoints with the smallest FIDs (on 1000 samples) encountered during training, and compute full FID and Inception scores on more samples from them. As shown by results in Table 1, NCSNv2 (w/ denoising) is able to significantly improve the FID scores of NCSN on both CIFAR-10 and CelebA, while bearing a slight loss of Inception scores on CIFAR-10. However, we note that Inception and FID scores have known issues [21, 22] and they should be interpreted with caution as they may not correlate with visual quality in the expected way. In particular, they can be sensitive to slight noise perturbations [23], as shown by the difference of

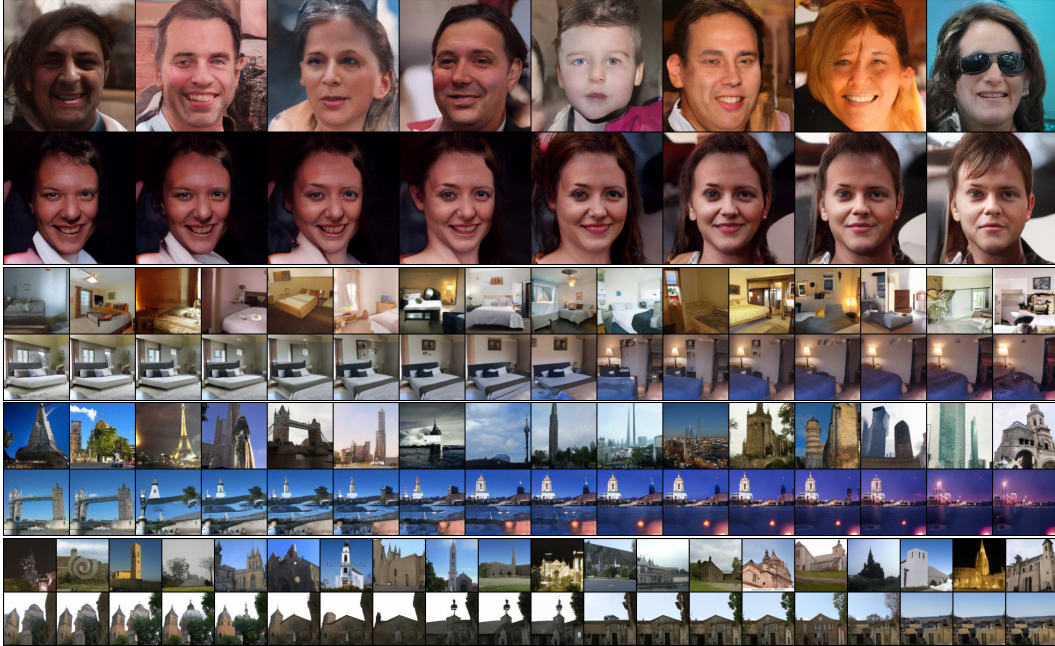

Figure 6: From top to bottom: FFHQ $256^2$, LSUN bedroom $128^2$, LSUN tower $128^2$, and LSUN church_outdoor $96^2$. Within each group of images: the first row shows uncurated samples from NCSNv2, and the second shows the interpolation results between the leftmost and rightmost samples with NCSNv2. You may zoom in to view more details.

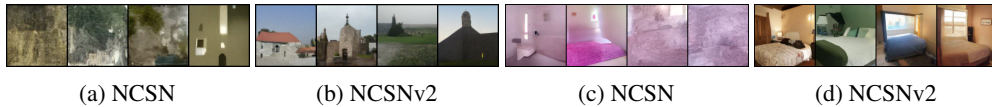

    (a) NCSN         (b) NCSNv2        (c) NCSN        (d) NCSNv2

Figure 7: NCSN vs. NCSNv2 samples on LSUN church_outdoor (a)(b) and LSUN bedroom (c)(d).

scores with and without denoising in Table 1. To verify that NCSNv2 indeed generates better images than NCSN, we provide additional uncurated samples in Appendix C.4 for visual comparison.

**Ablation studies:** We conduct ablation studies to isolate the contributions of different techniques. We partition all techniques into three groups: (i) Technique 5, (ii) Technique 1,2,4, and (iii) Technique 3, where different groups can be applied simultaneously. Technique 1,2 and 4 are grouped together because Technique 1 and 2 collectively determine the set of noise scales, and to sample from NCSNs trained with these noise scales we need Technique 4 to configure annealed Langevin dynamics properly. We test the performance of successively removing groups (iii), (ii), (i) from NCSNv2, and report results in Fig. 5 for sampling with denoising and in Fig. 9 (Appendix C.1) for sampling without denoising. All groups of techniques improve over the vanilla NCSN. Although the FID scores are not strictly increasing when removing (iii), (ii), and (i) progressively, we note that FIDs may not always correlate with sample quality well. In fact, we do observe decreasing sample quality by visual inspection (see Appendix C.4), and combining all techniques gives the best samples.

**Towards higher resolution:** The original NCSN only succeeds at generating images of low resolution. In fact, [1] only tested it on MNIST $28 \times 28$ and CelebA/CIFAR-10 $32 \times 32$. For slightly larger images such as CelebA $64 \times 64$, NCSN can generate images of consistent global structure, yet with strong color artifacts that are easily noticeable (see Fig. 4 and compare Fig. 10a with Fig. 10b). For images with resolutions beyond $96 \times 96$, NCSN will completely fail to produce samples with correct structure or color (see Fig. 7). All samples shown here are generated without the denoising step, but since $\sigma_L$ is very small, they are visually indistinguishable from ones with the denoising step.

By combining Technique 1–5, NCSNv2 can work on images of much higher resolution. Note that we directly calculated the noise scales for training NCSNs, and computed the step size for annealed

Langevin dynamics sampling without manual hyper-parameter tuning. The network architectures are the same across datasets, except that for ones with higher resolution we use more layers and more filters to ensure the receptive field and model capacity are large enough (see details in Appendix B.1). In Fig. 6 and 1, we show NCSNv2 is capable of generating high-fidelity image samples with resolutions ranging from $96 \times 96$ to $256 \times 256$. To show that this high sample quality is not a result of dataset memorization, we provide the loss curves for training/test, as well as nearest neighbors for samples in Appendix C.5. In addition, NCSNv2 can produce smooth interpolations between two given samples as in Fig. 6 (details in Appendix B.2), indicating the ability to learn generalizable image representations.

## 7 Conclusion

Motivated by both theoretical analyses and empirical observations, we propose a set of techniques to improve score-based generative models. Our techniques significantly improve the training and sampling processes, lead to better sample quality, and enable high-fidelity image generation at high resolutions. Although our techniques work well without manual tuning, we believe that the performance can be improved even more by fine-tuning various hyper-parameters. Future directions include theoretical understandings on the sample quality of score-based generative models, as well as alternative noise distributions to Gaussian perturbations.

## Broader Impact

Our work represents another step towards more powerful generative models. While we focused on images, it is quite likely that similar techniques could be applicable to other data modalities such as speech or behavioral data (in the context of imitation learning). Like other generative models that have been previously proposed, such as GANs and WaveNets, score models have a multitude of applications. Among many other applications, they could be used to synthesize new data automatically, detect anomalies and adversarial examples, and also improve results in key tasks such as semi-supervised learning and reinforcement learning. In turn, these techniques can have both positive and negative impacts on society, depending on the application. In particular, the models we trained on image datasets can be used to synthesize new images that are hard to distinguish from real ones by humans. Synthetic images from generative models have already been used to deceive humans in malicious ways. There are also positive uses of these technologies, for example in the arts and as a tool to aid design in engineering. We also note that our models have been trained on datasets that have biases (*e.g.*, CelebA is not gender-balanced), and the learned distribution is likely to have inherited them, in addition to others that are caused by the so-called inductive bias of models.

## Acknowledgments and Disclosure of Funding

The authors would like to thank Aditya Grover, Rui Shu and Shengjia Zhao for reviewing an early draft of this paper, as well as Gabby Wright and Sharon Zhou for resolving technical issues in computing $\text{HYPE}_\infty$ scores. This research was supported by NSF (#1651565, #1522054, #1733686), ONR (N00014-19-1-2145), AFOSR (FA9550-19-1-0024), and Amazon AWS.

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
