[Supplementary Material]

# A  Proofs

**Proposition 1.** *Let $\hat{p}_{\sigma_1}(\mathbf{x}) \triangleq \frac{1}{N}\sum_{i=1}^{N} p^{(i)}(\mathbf{x})$, where $p^{(i)}(\mathbf{x}) \triangleq \mathcal{N}(\mathbf{x} \mid \mathbf{x}^{(i)}, \sigma_1^2 I)$. With $r^{(i)}(\mathbf{x}) \triangleq$ $\frac{p^{(i)}(\mathbf{x})}{\sum_{k=1}^{N} p^{(k)}(\mathbf{x})}$, the score function is $\nabla_{\mathbf{x}} \log \hat{p}_{\sigma_1}(\mathbf{x}) = \sum_{i=1}^{N} r^{(i)}(\mathbf{x}) \nabla_{\mathbf{x}} \log p^{(i)}(\mathbf{x})$. Moreover,*

$$\mathbb{E}_{p^{(i)}(\mathbf{x})}[r^{(j)}(\mathbf{x})] \leq \frac{1}{2} \exp\left(-\frac{\left\|\mathbf{x}^{(i)} - \mathbf{x}^{(j)}\right\|_2^2}{8\sigma_1^2}\right). \tag{5}$$

*Proof.* According to the definition of $p_{\sigma_1}(\mathbf{x})$ and $r(\mathbf{x})$, we have

$$\nabla_{\mathbf{x}} \log \hat{p}_{\sigma_1}(\mathbf{x}) = \nabla_{\mathbf{x}} \log \left(\frac{1}{N}\sum_{i=1}^{N} p^{(i)}(\mathbf{x})\right) = \sum_{i=1}^{N} \frac{\nabla_{\mathbf{x}} p^{(i)}(\mathbf{x})}{\sum_{j=1}^{N} p^{(j)}(\mathbf{x})}$$

$$= \sum_{i=1}^{N} \frac{p^{(i)}(\mathbf{x}) \nabla_{\mathbf{x}} \log p^{(i)}(\mathbf{x})}{\sum_{j=1}^{N} p^{(j)}(\mathbf{x})}$$

$$= \sum_{i=1}^{N} r^{(i)}(\mathbf{x}) \nabla_{\mathbf{x}} \log p^{(i)}(\mathbf{x}).$$

Next, assuming $\mathbf{x} \in \mathbb{R}^D$, we have

$$\mathbb{E}_{p^{(i)}(\mathbf{x})}[r^{(j)}(\mathbf{x})] = \int \frac{p^{(i)}(\mathbf{x}) p^{(j)}(\mathbf{x})}{\sum_{k=1}^{N} p^{(k)}(\mathbf{x})} d\mathbf{x} \leq \int \frac{p^{(i)}(\mathbf{x}) p^{(j)}(\mathbf{x})}{p^{(i)}(\mathbf{x}) + p^{(j)}(\mathbf{x})} d\mathbf{x}$$

$$= \frac{1}{2} \int \frac{2}{\frac{1}{p^{(i)}(\mathbf{x})} + \frac{1}{p^{(j)}(\mathbf{x})}} d\mathbf{x} \overset{(1)}{\leq} \frac{1}{2} \int \sqrt{p^{(i)}(\mathbf{x}) p^{(j)}(\mathbf{x})} d\mathbf{x}$$

$$= \frac{1}{2} \frac{1}{(2\pi\sigma_1^2)^{D/2}} \int \exp\left(-\frac{1}{4\sigma_1^2}\left(\left\|\mathbf{x} - \mathbf{x}^{(i)}\right\|_2^2 + \left\|\mathbf{x} - \mathbf{x}^{(j)}\right\|_2^2\right)\right) d\mathbf{x}$$

$$= \frac{1}{2} \frac{1}{(2\pi\sigma_1^2)^{D/2}} \int \exp\left(-\frac{1}{4\sigma_1^2}\left(\left\|\mathbf{x} - \mathbf{x}^{(i)}\right\|_2^2 + \left\|\mathbf{x} - \mathbf{x}^{(i)} + \mathbf{x}^{(i)} - \mathbf{x}^{(j)}\right\|_2^2\right)\right) d\mathbf{x}$$

$$= \frac{1}{2} \frac{1}{(2\pi\sigma_1^2)^{D/2}} \int \exp\left\{-\frac{1}{2\sigma_1^2}\left(\left\|\mathbf{x} - \mathbf{x}^{(i)}\right\|_2^2 + (\mathbf{x} - \mathbf{x}^{(i)})^\mathsf{T}(\mathbf{x}^{(i)} - \mathbf{x}^{(j)}) + \frac{\left\|\mathbf{x}^{(i)} - \mathbf{x}^{(j)}\right\|_2^2}{2}\right)\right\} d\mathbf{x}$$

$$= \frac{1}{2} \frac{1}{(2\pi\sigma_1^2)^{D/2}} \int \exp\left\{-\frac{1}{2\sigma_1^2}\left(\left\|\mathbf{x} - \mathbf{x}^{(i)} + \frac{\mathbf{x}^{(i)} - \mathbf{x}^{(j)}}{2}\right\|_2^2 + \frac{\left\|\mathbf{x}^{(i)} - \mathbf{x}^{(j)}\right\|_2^2}{4}\right)\right\} d\mathbf{x}$$

$$= \frac{1}{2} \exp\left(-\frac{\left\|\mathbf{x}^{(i)} - \mathbf{x}^{(j)}\right\|_2^2}{8\sigma_1^2}\right) \frac{1}{(2\pi\sigma_1^2)^{D/2}} \int \exp\left\{-\frac{1}{2\sigma_1^2}\left(\left\|\mathbf{x} - \mathbf{x}^{(i)} + \frac{\mathbf{x}^{(i)} - \mathbf{x}^{(j)}}{2}\right\|_2^2\right)\right\} d\mathbf{x}$$

$$= \frac{1}{2} \exp\left(-\frac{\left\|\mathbf{x}^{(i)} - \mathbf{x}^{(j)}\right\|_2^2}{8\sigma_1^2}\right),$$

where $(1)$ is due to the geometric mean–harmonic mean inequality. $\qquad\square$

**Proposition 2.** *Let $\mathbf{x} \in \mathbb{R}^D \sim \mathcal{N}(\mathbf{0}, \sigma^2 I)$, and $r = \|\mathbf{x}\|_2$. We have*

$$p(r) = \frac{1}{2^{D/2-1}\Gamma(D/2)} \frac{r^{D-1}}{\sigma^D} \exp\left(-\frac{r^2}{2\sigma^2}\right) \quad \text{and} \quad r - \sqrt{D}\sigma \overset{d}{\to} \mathcal{N}(0, \sigma^2/2) \text{ when } D \to \infty.$$

*Proof.* Since $\mathbf{x} \sim \mathcal{N}(\mathbf{0}, \sigma^2 I)$, we have $s \triangleq \|\mathbf{x}\|_2^2 / \sigma^2 \sim \chi_D^2$, *i.e.*,

$$p_s(s) = \frac{1}{2^{D/2}\Gamma(D/2)} s^{D/2-1} e^{-s/2}.$$

Because $r = \|\mathbf{x}\|_2 = \sigma\sqrt{s}$, we can use the change of variables formula to get

$$p(r) = \frac{2r}{\sigma^2} p_s(s) = \frac{2r}{\sigma^2} p_s\left(\frac{r^2}{\sigma^2}\right) = \frac{1}{2^{D/2-1}\Gamma(D/2)} \frac{r^{D-1}}{\sigma^D} \exp\left(-\frac{r^2}{2\sigma^2}\right),$$

which proves our first result. Next, we notice that if $x \sim \mathcal{N}(0, \sigma^2)$, we have $x^2/\sigma^2 \sim \chi_1^2$ and thus $\mathbb{E}[x] = \sigma^2$, $\text{Var}[x] = 2\sigma^4$. As a result, if $x_1, x_2, \cdots, x_D \overset{\text{i.i.d.}}{\sim} \mathcal{N}(0, \sigma^2)$, the law of large numbers and the central limit theorem will imply that as $D \to \infty$, both of the following hold:

$$\frac{x_1^2 + x_2^2 + \cdots + x_D^2}{D} \overset{p}{\to} \sigma^2$$

$$\sqrt{D}\left(\frac{x_1^2 + x_2^2 + \cdots + x_D^2}{D} - \sigma^2\right) \overset{d}{\to} \mathcal{N}(0, 2\sigma^4).$$

Equivalently,

$$\sqrt{D}\left(\frac{r^2}{D} - \sigma^2\right) \overset{d}{\to} \mathcal{N}(0, 2\sigma^4).$$

Applying the delta method, we obtain

$$\sqrt{D}\left(\frac{r}{\sqrt{D}} - \sigma\right) \overset{d}{\to} \mathcal{N}(0, \sigma^2/2),$$

and therefore $r - \sqrt{D}\sigma \overset{d}{\to} \mathcal{N}(0, \sigma^2/2)$. $\qquad\qquad\square$

**Proposition 3.** *Let* $\gamma = \frac{\sigma_{i-1}}{\sigma_i}$. *For* $\alpha = \epsilon \cdot \frac{\sigma_i^2}{\sigma_L^2}$ *(as in Algorithm 1), we have* $\mathbf{x}_T \sim \mathcal{N}(\mathbf{0}, s_T^2 I)$, *where*

$$\frac{s_T^2}{\sigma_i^2} = \left(1 - \frac{\epsilon}{\sigma_L^2}\right)^{2T}\left(\gamma^2 - \frac{2\epsilon}{\sigma_L^2 - \sigma_L^2\left(1 - \frac{\epsilon}{\sigma_L^2}\right)^2}\right) + \frac{2\epsilon}{\sigma_L^2 - \sigma_L^2\left(1 - \frac{\epsilon}{\sigma_L^2}\right)^2}. \qquad (6)$$

*Proof.* First, the conditions we know are

$$\mathbf{x}_0 \sim p_{\sigma_{i-1}}(\mathbf{x}) = \mathcal{N}(\mathbf{0}, \sigma_{i-1}^2 I),$$

$$\mathbf{x}_{t+1} \leftarrow \mathbf{x}_t + \alpha \nabla_{\mathbf{x}} \log p_{\sigma_i}(\mathbf{x}_t) + \sqrt{2\alpha}\mathbf{z}_t = \mathbf{x}_t - \alpha\frac{\mathbf{x}_t}{\sigma_i^2} + \sqrt{2\alpha}\mathbf{z}_t,$$

where $\mathbf{z}_t \sim \mathcal{N}(\mathbf{0}, I)$. Therefore, the variance of $\mathbf{x}_t$ satisfies

$$\text{Var}[\mathbf{x}_t] = \begin{cases} \sigma_{i-1}^2 I & \text{if } t = 0 \\ \left(1 - \frac{\alpha}{\sigma_i^2}\right)^2 \text{Var}[\mathbf{x}_{t-1}] + 2\alpha I & \text{otherwise.} \end{cases}$$

Now let $\mathbf{v} \triangleq \frac{2\alpha}{1 - \left(1 - \frac{\alpha}{\sigma_i^2}\right)^2} I$, we have

$$\text{Var}[\mathbf{x}_t] - \mathbf{v} = \left(1 - \frac{\alpha}{\sigma_i^2}\right)^2 (\text{Var}[\mathbf{x}_{t-1}] - \mathbf{v}).$$

Therefore,

$$\text{Var}[\mathbf{x}_T] - \mathbf{v} = \left(1 - \frac{\alpha}{\sigma_i^2}\right)^{2T} (\text{Var}[\mathbf{x}_0] - \mathbf{v})$$

$$\implies \text{Var}[\mathbf{x}_T] = \left(1 - \frac{\alpha}{\sigma_i^2}\right)^{2T} (\text{Var}[\mathbf{x}_0] - \mathbf{v}) + \mathbf{v}$$

$$\implies s_T^2 = \left(1 - \frac{\alpha}{\sigma_i^2}\right)^{2T}\left(\sigma_{i-1}^2 - \frac{2\alpha}{1 - \left(1 - \frac{\alpha}{\sigma_i^2}\right)^2}\right) + \frac{2\alpha}{1 - \left(1 - \frac{\alpha}{\sigma_i^2}\right)^2}. \qquad (7)$$

Substituting $\epsilon\, \sigma_i^2/\sigma_L^2$ for $\alpha$ in Eq. (7), we immediately obtain Eq. (6). $\qquad\qquad\square$

# B Experimental details

## B.1 Network architectures and hyperparameters

The original NCSN in [1] uses a network structure based on RefineNet [24]—a classical architecture for semantic segmentation. There are three major modifications to the original RefineNet in NCSN: (i) adding an enhanced version of conditional instance normalization (designed in [1] and named CondInstanceNorm++) for every convolutional layer; (ii) replacing max pooling with average pooling in RefineNet blocks; and (iii) using dilated convolutions in the ResNet backend of RefineNet. We use exactly the same architecture for NCSN experiments, but for NCSNv2 or any other architecture implementing Technique 3, we apply the following modifications: (i) setting the number of classes in CondInstanceNorm++ to 1 (which we name as InstanceNorm++); (ii) changing average pooling back to max pooling; and (iii) removing all normalization layers in RefineNet blocks. Here (ii) and (iii) do not affect the results much, but they are included because we hope to minimize the number of unnecessary changes to the standard RefineNet architecture (the original RefineNet blocks in [24] use max pooling and have no normalization layers). We name a ResNet block (with InstanceNorm++ instead of BatchNorm) "ResBlock", and a RefineNet block "RefineBlock". When CondInstanceNorm++ is added, we name them "CondResBlock" and "CondRefineBlock" respectively. We use the ELU activation function [25] throughout all architectures.

To ensure sufficient capacity and receptive fields, the network structures for images of different resolutions have different numbers of layers and filters. We summarize the architectures in Table 2 and Table 3.

Table 2: The architectures of NCSN for images of various resolutions.

(a) NCSN $32^2$–$64^2$

| |
|---|
| 3x3 Conv2D, 128 |
| CondResBlock, 128 |
| CondResBlock, 128 |
| CondResBlock down, 256 |
| CondResBlock, 256 |
| CondResBlock down, 256 dilation 2 |
| CondResBlock, 256 dilation 2 |
| CondResBlock down, 256 dilation 4 |
| CondResBlock, 256 dilation 4 |
| CondRefineBlock, 256 |
| CondRefineBlock, 256 |
| CondRefineBlock, 128 |
| CondRefineBlock, 128 |
| 3x3 Conv2D, 3 |

(b) NCSN $96^2$–$128^2$

| |
|---|
| 3x3 Conv2D, 128 |
| CondResBlock, 128 |
| CondResBlock, 128 |
| CondResBlock down, 256 |
| CondResBlock, 256 |
| CondResBlock down, 256 |
| CondResBlock, 256 |
| CondResBlock down, 512 dilation 2 |
| CondResBlock, 512 dilation 2 |
| CondResBlock down, 512 dilation 4 |
| CondResBlock, 512 dilation 4 |
| CondRefineBlock, 512 |
| CondRefineBlock, 256 |
| CondRefineBlock, 256 |
| CondRefineBlock, 128 |
| CondRefineBlock, 128 |
| 3x3 Conv2D, 3 |

Table 3: The architectures of NCSNv2 for images of various resolutions.

(a) NCSNv2 $32^2$–$64^2$

| |
| --- |
| 3x3 Conv2D, 128 |
| ResBlock, 128 |
| ResBlock, 128 |
| ResBlock down, 256 |
| ResBlock, 256 |
| ResBlock down, 256 dilation 2 |
| ResBlock, 256 dilation 2 |
| ResBlock down, 256 dilation 4 |
| ResBlock, 256 dilation 4 |
| RefineBlock, 256 |
| RefineBlock, 256 |
| RefineBlock, 128 |
| RefineBlock, 128 |
| 3x3 Conv2D, 3 |

(b) NCSNv2 $96^2$–$128^2$

| |
| --- |
| 3x3 Conv2D, 128 |
| ResBlock, 128 |
| ResBlock, 128 |
| ResBlock down, 256 |
| ResBlock, 256 |
| ResBlock down, 256 |
| ResBlock, 256 |
| ResBlock down, 512 dilation 2 |
| ResBlock, 512 dilation 2 |
| ResBlock down, 512 dilation 4 |
| ResBlock, 512 dilation 4 |
| RefineBlock, 512 |
| RefineBlock, 256 |
| RefineBlock, 256 |
| RefineBlock, 128 |
| RefineBlock, 128 |
| 3x3 Conv2D, 3 |

(c) NCSNv2 $256^2$

| |
| --- |
| 3x3 Conv2D, 128 |
| ResBlock, 128 |
| ResBlock, 128 |
| ResBlock down, 256 |
| ResBlock, 256 |
| ResBlock down, 256 |
| ResBlock, 256 |
| ResBlock down, 256 |
| ResBlock, 256 |
| ResBlock down, 512 dilation 2 |
| ResBlock, 512 dilation 2 |
| ResBlock down, 512 dilation 4 |
| ResBlock, 512 dilation 4 |
| RefineBlock, 512 |
| RefineBlock, 256 |
| RefineBlock, 256 |
| RefineBlock, 256 |
| RefineBlock, 128 |
| RefineBlock, 128 |
| 3x3 Conv2D, 3 |

We use the Adam optimizer [26] for all models. When Technique 3 is not in effect, we choose the learning rate 0.001; otherwise we use a learning rate 0.0001 to avoid loss explosion. We set the $\epsilon$ parameter of Adam to $10^{-3}$ for FFHQ and $10^{-8}$ otherwise. We provide other hyperparameters in Table 4, where $\sigma_1$, $L$, $T$, and $\epsilon$ of NCSNv2 are all chosen in accordance with our proposed techniques. When the number of training data is larger than 60000, we randomly sample 10000 of them and compute the maximum pairwise distance, which is set as $\sigma_1$ for NCSNv2.

Table 4: Hyperparameters of NCSN/NCSNv2. The latter is configured according to Technique 1–4. $\sigma_1$ and $L$ determine the set of noise levels. $T$ and $\epsilon$ are parameters of annealed Langevin dynamics.

| Model | Dataset | $\sigma_1$ | $L$ | $T$ | $\epsilon$ | Batch size | Training iterations |
| --- | --- | --- | --- | --- | --- | --- | --- |
| NCSN | CIFAR-10 $32^2$ | 1 | 10 | 100 | 2e-5 | 128 | 300k |
| NCSN | CelebA $64^2$ | 1 | 10 | 100 | 2e-5 | 128 | 210k |
| NCSN | LSUN church_outdoor $96^2$ | 1 | 10 | 100 | 2e-5 | 128 | 200k |
| NCSN | LSUN bedroom $128^2$ | 1 | 10 | 100 | 2e-5 | 64 | 150k |
| NCSNv2 | CIFAR-10 $32^2$ | 50 | 232 | 5 | 6.2e-6 | 128 | 300k |
| NCSNv2 | CelebA $64^2$ | 90 | 500 | 5 | 3.3e-6 | 128 | 210k |
| NCSNv2 | LSUN church_outdoor $96^2$ | 140 | 788 | 4 | 4.9e-6 | 128 | 200k |
| NCSNv2 | LSUN bedroom/tower $128^2$ | 190 | 1086 | 3 | 1.8e-6 | 128 | 150k |
| NCSNv2 | FFHQ $256^2$ | 348 | 2311 | 3 | 0.9e-7 | 32 | 80k |

## B.2 Additional settings

**Datasets:** We use the following datasets in our experiments: CIFAR-10 [2], CelebA [16], LSUN [27], and FFHQ [28]. CIFAR-10 contains 50000 training images and 10000 test images, all of resolution $32 \times 32$. CelebA contains 162770 training images and 19962 test images with various resolutions. For preprocessing, we first center crop them to size $140 \times 140$, and then resize them to $64 \times 64$. We choose the church_outdoor, bedroom and tower categories in the LSUN dataset. They contain 126227, 3033042, and 708264 training images respectively, and all have 300 validation images. For preprocessing, we first resize them so that the smallest dimension of images is 96 (for church_outdoor) or 128 (for bedroom and tower), and then center crop them to equalize their lengths and heights. Finally, the FFHQ dataset consists of 70000 high-quality facial images at resolution $1024 \times 1024$. We resize them to $256 \times 256$ in our experiments. Because FFHQ does not have an official test dataset, we randomly select 63000 images for training and the remaining 7000 as the test dataset. In addition, we apply random horizontal flip as data augmentation in all cases.

**Metrics:** We use FID [13] and $\text{HYPE}_\infty$ [29] scores for quantitative comparison of results. When computing FIDs on CIFAR-10 $32 \times 32$, we measure the distance between the statistics of samples and training data. When computing FIDs on CelebA $64 \times 64$, we follow the settings in [30] where the distance is measured between 10000 samples and the test dataset. We use the official website https://hype.stanford.edu for computing $\text{HYPE}_\infty$ scores. Regarding model selection, we follow the settings in [1], where we compute FID scores on 1000 samples every 5000 training iterations and choose the checkpoint with the smallest FID for computing both full FID scores (with more samples from it) and the $\text{HYPE}_\infty$ scores.

**Training:** We use the Adam [26] optimizer with default hyperparameters. The learning rates and batch sizes are provided in Appendix B.1 and Table 4. We observe that for images at resolution $128 \times 128$ or $256 \times 256$, training can be unstable when the loss is near convergence. We note, however, this is a well-known problem of the Adam optimizer, and can be mitigated by techniques such as AMSGrad [31]. We trained all models on Nvidia Tesla V100 GPUs.

**Settings for Section 3.3:** The loss curves in Fig. 3 are results of two settings: (i) Technique 1, 2, 4 and 5 are in effect, but the model architecture is the same as the original NCSN (*i.e.*, Table 2a); and (ii) all techniques are in effect, *i.e.*, the model is the same as NCSNv2 depicted in Table 3a. We apply EMA with momentum 0.9 to smooth the curves in Fig. 3. We observe that despite being simpler to implement, the new noise conditioning method proposed in Technique 3 performs as well as the original and arguably more complex one in [1] in terms of the training loss. See the ablation studies in Section 6 and Appendix C.4 for additional results.

**Interpolation:** We can interpolate between two different samples from NCSN/NCSNv2 via interpolating the Gaussian random noise injected by annealed Langevin dynamics. Specifically, suppose we have a total of $L$ noise levels, and for each noise level we run $T$ steps of Langevin dynamics. Let $\{\mathbf{z}_{ij}\}_{1 \le i \le L, 1 \le j \le T} \triangleq \{\mathbf{z}_{11}, \mathbf{z}_{12}, \cdots, \mathbf{z}_{1T}, \mathbf{z}_{21}, \mathbf{z}_{22}, \cdots, \mathbf{z}_{2T}, \cdots, \mathbf{z}_{L1}, \mathbf{z}_{L2}, \cdots, \mathbf{z}_{LT}\}$ denote the set of all Gaussian noise used in this procedure, where $\mathbf{z}_{ij}$ is the noise injected at the $j$-th iteration of Langevin dynamics corresponding to the $i$-th noise level. Next, suppose we have two samples $\mathbf{x}^{(1)}$ and $\mathbf{x}^{(2)}$ with the same initialization $\mathbf{x}_0$, and denote the corresponding set of Gaussian noise as $\{\mathbf{z}_{ij}^{(1)}\}_{1 \le i \le L, 1 \le j \le T}$ and $\{\mathbf{z}_{ij}^{(2)}\}_{1 \le i \le L, 1 \le j \le T}$ respectively. We can generate $N$ interpolated samples between $\mathbf{x}^{(1)}$ and $\mathbf{x}^{(2)}$, where for the $k$-th interpolated sample we use Gaussian noise $\{\cos\left(\frac{k\pi}{2(N+1)}\right)\mathbf{z}_{ij}^{(1)} + \sin\left(\frac{k\pi}{2(N+1)}\right)\mathbf{z}_{ij}^{(2)}\}_{1 \le i \le L, 1 \le j \le T}$ and initialization $\mathbf{x}_0$.

## C Additional experimental results

### C.1 Additional results without the denoising step

We further demonstrate the stabilizing effect of EMA in Fig. 8, where FIDs are computed without the denoising step. As indicated by Figs. 4 and 8, EMA can stabilize training and remove sample artifacts regardless of whether denoising is used or not.

FID scores should be interpreted with caution because they may not align well with human judgement. For example, the samples from NCSNv2 as demonstrated in Fig. 10b have an FID score of 28.9 (without denoising), worse than NCSN (Fig. 10a) whose FID is 26.9 (without denoising), but arguably

Figure 8: FIDs and color artifacts over the course of training (best viewed in color). The FIDs of NCSN have much higher volatility compared to NCSN with EMA. Samples from the vanilla NCSN often have obvious color shifts. All FIDs are computed without the denoising step.

(a) CIFAR-10 FIDs     (b) CelebA FIDs

Figure 9: FIDs for different groups of techniques. Subscripts of "NCSN" are IDs of techniques in effect. "NCSNv2" uses all techniques. Results are computed without the denoising step.

(a) NCSN     (b) NCSNv2

Figure 10: Uncurated samples from NCSN (a) and NCSNv2 (b) on CelebA $64 \times 64$.

produce much more visually appealing samples. To investigate whether FID scores align well with human ratings, we use the $\text{HYPE}_\infty$ [29] score (higher is better), a metric of sample quality based on human evaluation, to compare the two models that generated samples in Figs. 10a and 10b. We provide full results in Table 5, where all numbers except those for NCSN and NCSNv2 are directly taken from [29]. As Table 5 shows, our NCSNv2 achieves 37.3 on CelebA $64 \times 64$ which is comparable to ProgressiveGAN [32], whereas NCSN achieves 19.8. This is completely different from the ranking indicated by FIDs.

Table 5: $\text{HYPE}_\infty$ scores on CelebA $64 \times 64$. *With truncation tricks.

| Model | $\text{HYPE}_\infty$(%) | Fakes Error(%) | Reals Error(%) | Std. |
|---|---|---|---|---|
| StyleGAN* [28] | 50.7 | 62.2 | 39.3 | 1.3 |
| ProgressiveGAN [32] | 40.3 | 46.2 | 34.4 | 0.9 |
| BEGAN [33] | 10 | 6.2 | 13.8 | 1.6 |
| WGAN-GP [19] | 3.8 | 1.7 | 5.9 | 0.6 |
| NCSN | 19.8 | 22.3 | 17.3 | 0.4 |
| NCSNv2 | 37.3 | 49.8 | 24.8 | 0.5 |

* with truncation tricks

Finally, we provide ablation results without the denoising step in Fig. 9. It is qualitatively similar to Fig. 5 where results are computed with denoising.

## C.2 Training and sampling speed

In Table 6, we provide the time cost for training and sampling from NCSNv2 models on various datasets considered in our experiments.

Table 6: Training and sampling speed of NCSNv2 on various datasets.

| Dataset | Device | Sampling time | Training time |
|---------|--------|---------------|---------------|
| CIFAR-10 | 2x V100 | 2 min | 22 h |
| CelebA | 4x V100 | 7 min | 29 h |
| Church | 8x V100 | 17 min | 52 h |
| Bedroom | 8x V100 | 19 min | 52 h |
| Tower | 8x V100 | 19 min | 52 h |
| FFHQ | 8x V100 | 50 min | 41 h |

## C.3 Color shifts

(a) NCSN (Iter. = 50k)　(b) NCSN (Iter. = 100k)　(c) NCSN (Iter. = 200k)

(d) NCSN w/ EMA (Iter. = 50k)　(e) NCSN w/ EMA (Iter. = 100k)　(f) NCSN w/ EMA (Iter. = 200k)

Figure 11: EMA reduces undesirable color shifts on CIFAR-10. We show samples from NCSN and NCSN with EMA at the 50k/100k/200k-th training iteration.

(a) NCSN (Iter. = 50k)     (b) NCSN (Iter. = 100k)     (c) NCSN (Iter. = 150k)

(d) NCSN w/ EMA (Iter. = 50k)   (e) NCSN w/ EMA (Iter. = 100k)   (f) NCSN w/ EMA (Iter. = 150k)

Figure 12: EMA reduces undesirable color shifts on CelebA-10. We show samples from NCSN and NCSN with EMA at the 50k/100k/150k-th training iteration.

## C.4 Additional results on ablation studies

As discussed in Section 6, we partition all techniques into three groups: (i) Technique 5, (ii) Technique 1,2,4, and (iii) Technique 3, and investigate the performance of models after successively removing (iii), (ii), and (i) from NCSNv2. Aside from the FID curves in Figs. 5 and 9, we also provide samples from different models for visual inspection in Figs. 13 and 14. To generate these samples, we compute the FID scores on 1000 samples every 5000 training iterations for each considered model, and sample from the checkpoint of the smallest FID (the same setting as in [1]). From samples in Figs. 13 and 14, we easily observe that removing any group of techniques leads to worse samples.

(a) NCSN on CIFAR-10

(b) NCSN on CelebA

(c) NCSN$_5$ on CIFAR-10

(d) NCSN$_5$ on CelebA

(e) NCSN$_{1,2,4,5}$ on CIFAR-10

(f) NCSN$_{1,2,4,5}$ on CelebA

(g) NCSNv2 on CIFAR-10

(h) NCSNv2 on CelebA

Figure 13: Samples from models with different groups of techniques applied. NCSN is the original model in [1] and does not use any of the newly proposed techniques. Subscripts of "NCSN" denote the IDs of techniques in effect. NCSN$_5$ only applies EMA. NCSN$_{1,2,4,5}$ applies both EMA and technique group (ii). NCSNv2 is the result of all techniques combined. Checkpoints are selected according to the lowest FID (with denoising) over the course of training.

(a) NCSN on CIFAR-10

(b) NCSN on CelebA

(c) NCSN$_5$ on CIFAR-10

(d) NCSN$_5$ on CelebA

(e) NCSN$_{1,2,4,5}$ on CIFAR-10

(f) NCSN$_{1,2,4,5}$ on CelebA

(g) NCSNv2 on CIFAR-10

(h) NCSNv2 on CelebA

Figure 14: Samples from models with different groups of techniques applied. NCSN is the original model in [1] and does not use any of the newly proposed techniques. Subscripts of "NCSN" denote the IDs of techniques in effect. NCSN$_5$ only applies EMA. NCSN$_{1,2,4,5}$ applies both EMA and technique group (ii). NCSNv2 is the result of all techniques combined. Checkpoints are selected according to the lowest FID (without denoising) over the course of training.

## C.5 Generalization

### C.5.1 Loss curves

First, we demonstrate that our NCSNv2 does not overfit to the training dataset by showing the curves of training/test loss in Fig. 15. Since the loss on the test dataset is always close to the loss on the training dataset during the course of training, this indicates that our model does not simply memorize training data.

Figure 15: Training vs. test loss curves of NCSNv2.

### C.5.2 Nearest neighbors

Starting from this section, all samples are from NCSNv2 at the last training iteration. For each generated sample, we show the nearest neighbors from the training dataset, measured by $\ell_2$ distance in the feature space of a pre-trained InceptionV3 network. Since we apply random horizontal flip when training, we also take this into consideration when computing nearest neighbors, so that we can detect cases in which NCSNv2 memorizes a flipped training data point.

Figure 16: Nearest neighbors on CIFAR-10. NCSNv2 samples are on the left side of the red vertical line. Corresponding nearest neighbors are on the right side in the same row.

Figure 17: Nearest neighbors on CelebA $64 \times 64$.

Figure 18: Nearest neighbors on LSUN church_outdoor $96 \times 96$.

Figure 19: Nearest neighbors on FFHQ $256 \times 256$.

### C.5.3 Additional interpolation results

We generate samples from NCSNv2 and interpolate between them using the method described in Appendix B.2.

Figure 20: NCSNv2 interpolation results on CelebA $64 \times 64$.

Figure 21: NCSNv2 interpolation results on LSUN church_outdoor $96 \times 96$.

Figure 22: NCSNv2 interpolation results on LSUN bedroom $128 \times 128$.

Figure 23: NCSNv2 interpolation results on LSUN tower $128 \times 128$.

Figure 24: NCSNv2 interpolation results on FFHQ $256 \times 256$.

## C.6 Additional uncurated samples

Figure 25: Uncurated CIFAR-10 $32 \times 32$ samples from NCSNv2.

Figure 26: Uncurated CelebA $64 \times 64$ samples from NCSNv2.

Figure 27: Uncurated LSUN church_outdoor 96 × 96 samples from NCSNv2.

Figure 28: Uncurated LSUN bedroom $128 \times 128$ samples from NCSNv2.

Figure 29: Uncurated LSUN tower $128 \times 128$ samples from NCSNv2.

Figure 30: Uncurated FFHQ $256 \times 256$ samples from NCSNv2.