[Reviews · NeurIPS 2020]

Review 1

Summary and Contributions: This paper analyzes the main challenges for training score based generative models in practice: the choice of noise level, the large number of noise level and the training stability. The authors clearly analyze the reasons for these issues and provide a set of practically useful techniques to improve the performance and stability of score based generative models. This paper is well written and the experimental results are sufficient. The ablation study for these techniques and and results on large scale dataset is convincing.

Strengths: This paper is well written. This paper is theoretically sound. The experimental results are sufficient and convincing.

Weaknesses: I have some minor concerns about the experimental results as follows: As shown in Figure 1, the generated samples in LSUN tower dataset is very similar to the real samples, like the Eiffel Tower. Does it mean that the NCSNv2 actually overfits to the training data? I think that demonstrating a pair data of the generated samples and its most similar real samples of the training data under certain norm, e.g., L2 norm, is useful to address my concern. For Figure 4, it seems that the best FID achieved by NCSN is better than the EMA version. Indeed the EMA can stabilize the training, but I wonder whether it will have a negative effect on the performance. Since the number of noise level is very large (e.g., over 2k for FFHQ dataset), is it still necessary to quantize the noise level instead of simply random sampling it from certain distribution such that p(\sigma \in [\sigma_{i-1}, \sigma_i]) \approx 1/L. Here $\sigma_i$ is the i-th item of the geometric sequence defined in Technique 2.

Correctness: This paper is theoretically sound.

Clarity: This paper is well written.

Relation to Prior Work: Yes

Reproducibility: Yes

Additional Feedback:


Review 2

Summary and Contributions: Much like the "Improved Techniques for Training GANs" of four years ago, the authors of this paper propose a set of five techniques to improve the quality of the recently-introduced Noise-Conditional Score Networks. Three of the five techniques are more principled design choices for the three main hyperparameters of annealed Langevin dynamics: the initial noise parameter (technique 1), the noise sequence (technique 2), the step size/number of steps for a given noise (technique 4). The other two techniques are architectural: EMA in the network (technique 5), and simplification of the noise-conditional score network (technique 3). Combining these techniques allows the model to learn 256x256 FFHQ faces, and improves stability and sample quality of 32x32 and 64x64 images.

Strengths: Certainly, the ability to train on higher-resolution images is a main selling point of the paper. Being able to scale to 256x256 images (even if they lag in sample quality compared to other model classes) is fairly impressive. Also, even though the hyperparameter choices are not backed by "rigorous" analysis, I did like that the authors provided at least some theoretical justification, which seemingly improved results in the experimental section. Finally, the samples from NCSNv2 certainly look better than NCSN on CelebA 64x64 and CIFAR-10, even if the FID scores do not reflect an improvement.

Weaknesses: The experimental section does not quite have the experiments I was hoping for. I was hoping to understand *which* techniques were important to scale the model to higher-resolution images. I know that using all techniques are needed for the model to learn LSUN images, but my suspicion is that only a subset of them (perhaps only EMA) is needed for regular NCSN to learn high-resolution images. Better understanding of which techniques are important would help explain what is needed for scaling the model. Moreover, for the ablation experiments provided, it does not seem that all 5 techniques are need for all datasets. Fig 5 seems to show that the simplified network is not needed on CelebA 64x64 as FID is better using the original network. In fact, the samples in Figure 11f and 11g seem to show relatively similar sample quality. Between Figure 5, Figure 11, and Appendix C.2, it seems that EMA both improves training stability and reduces the color shift problem. The cost for this, it seems, is that "peak" FID is no longer as good as that for the original NCSN. So techniques 1, 2, and 4 (and depending on the dataset, technique 3), help improve FID. Unfortunately, while the techniques reduce the gap in performance between NCSNv2 and NCSN on CelebA/CIFAR10, NCSNv2 does not have better FIDs than NCSNv1. If other techniques, without EMA, can help solve the training stability/color shift problem, I'd love to take a look at the experiments. One thing not mentioned in the main paper -- and not fully explored in the experiments -- is that the RefineNet architecture is changed between the previous version and current one. I'm assuming the NCSN reproduction experiments (Table 5) use the old version of the architecture, but the NCSN ablation experiments use the newer one? How do results change between the different architectures? Finally, please do not oversell the paper. Words such as "effortlessly" and "unprecedented" detract from the paper. First, I'm not exactly sure to what "effort" refers. To researcher "effort"? To computational "effort"? Something else? And "unprecedented" only highlights how limited the rest of the claim is (that it's for score-based generative models). The results are good on their own!

Correctness: Apart from the concerns highlighted above, there seems to be larger issues with evaluating NCSNs (both v1 and v2) using GAN-based metrics. As noted in [1] and [2], FID (and Inception Score) seem to unfairly penalize non-GAN models (in that if there is a GAN model and non-GAN model of similar sample quality, FID/IS will be worse for the non-GAN model). This paper seems to suggest something different: even for models within the the same model class, FID is not representative of sample quality. Is this also true of Inception Score (which is included in the original NCSN paper, but not this one)? And if we have limited faith in FID, how seriously should be take Figure 5? And while the Hype_{\infty} results on CelebA 64x64 seem quite promising (and are more indicative of true sample quality), isn't this score only based on 50 samples (each) from the model and data? Did you use curated samples from NCSN (and NCSN-v2)? [1] Ali Razavi, Aaron van den Oord, and Oriol Vinyals. "Generating diverse high-fidelity images with VQ-VAE-2." arXiv preprint arXiv:1906.00446, 2019. [2] S. Ravuri and O. Vinyals, “Classification accuracy score for conditional generative models,” in Proc. Advances Neural Inf. Process. Syst., 2019.

Clarity: The paper is very clearly written. I only found one typo line 256 bett -> better

Relation to Prior Work: I think there are some references missing. Technique 5 is the same one used as Progressive GAN, so it should be stated clearly that the method is at least partially cadged from that paper.

Reproducibility: Yes

Additional Feedback: - Sampling, procedurally, requires ~1000 forward passes through a RefineNet, which seems quite expensive. How long does it take? Would it be possible to include this information in the paper? - Why is it that you include the authors' names (Song and Ermon) when you cite [1], but do not include the author names for any other references? - For Hype_{\infty}, wouldn't you want scores as close to 50 as possible (which is chance performance with the data), rather than 100 (which is hyperreal)? - What are "fakes" and "reals error" in Table 6? - In Table 1, you should state that the GAN results were taken from the original Hype paper. ### Post Rebuttal Update ### In general, I am happy with the rebuttal and increased my score to 7. Two points, however, remain unresolved: 1) The baseline Hype-infinity scores in Table 2 (for StyleGAN, ProgressiveGAN, BEGAN, and WGAN-GP), are taken from the original Zhou paper. Those scores were obtained using 50 real and 50 generated samples, while the NCSN scores were obtained using 2000 per the authors' feedback. As a result, these results are not directly comparable. 2) I should note that the "surprising fact of FID computation" in the authors' feedback -- that small amounts of noise greatly hurts FID computation -- was previously observed by the VQ-VAE-2 authors (reference [2] above). To quote them: "we observe that the inception classifier is quite sensitive to slight blurriness". Running a "Langevin step" without Gaussian noise, however, is novel.


Review 3

Summary and Contributions: Existing noise-conditional score network (NCSN) learn the score function (i.e., gradient of log density) by optimizing multi-scale denoising score matching objectives, where the data perturbing noises are a geometric sequence: sigma={1.0,...,0.01}. However, NCSN only works for low dimensional image datasets (i.e., 32x32 resolution). This paper proposes several training techniques that scales NCSN to higher dimensional image data (up to 256x256). The main contributions are theoretical analysis of how to set the largest perturbing noise sigma, number of noises, the learning rate. Regarding the experiment results, NCSNv2 improves NCSN qualitatively from the provided samples. Quantitatively speaking, NCSNv2 achieves a higher HYPE_infty score than NCSN. However, NCSNv2 does not outperform NCSN when comparing the FID score on CIFAR-10 and CelebA, as indicated in Table 5 of the Appendix.

Strengths: (1) Visually speaking, NCSNv2 has good generation results on high-dimensional images. (2) Solid analysis in Section 3.1 regarding the setting of Initial noise level.

Weaknesses: (1) Oversimplified assumption in Section 3.2 about the datasets only consists of a single data point. For any real-world high-dimensional, multi-modal data distributions, this assumption will not hold. This weakens the Technique 2 about how to set the total number of noises. A similar concern holds for Proposition 3. (2) Regarding the evaluation metric in image generation, people typically compare Inception Score and FID score, perhaps also the recently popular Precision/Recall. Nevertheless, NCSNv2 seems to have a worse FID score compared to the original NCSN (see Table 5, Appendix). This empirical result also conflicts with the conclusion drawn from Figure 5.

Correctness: Most of the theoretical analysis seems technically sound, except the oversimplified assumption part.

Clarity: The presentation and writing of this paper is generally clear.

Relation to Prior Work: The related work is sufficiently discussed.

Reproducibility: Yes

Additional Feedback: (1) Theoretically speaking, the optimal NCSN learns the score function of the kernel density estimator, for all noise levels. An alternative way is to directly use the multiscale KDE to conduct annealed SGLD, which dont even require training at all. This seems to question the motivation of NCSN? What is the FID score of KDE? It seems that the KDE performed well from Figure 2. (2) I felt like the most important factor is Technique 1, 2, 4. A more interesting ablation study would be running the original NCSN using Technique 1, 2, 4, and see if the FID score degrades or not. (3) As mentioned in the previous part, the quantitative evaluation of FID score seems somehow inconsistent in this paper. Figure 5 shows NCSNv2 has a better FID score on CIFAR-10 and CelebA. However, from Table 5 of Appendix, NCSNv2 has a worse FID score compared to the original NCSN. What caused this discrepancy? It would be great if the author can also present the Inception score or even Precision/Recall metric, for double check the conclusion,


Review 4

Summary and Contributions: The paper describes several rules of thumb for choosing parameters for training score-based generative image models. The rules are derived from first-principles analyses. Together, the contributions significantly increase the resolution and quality obtainable from score-based generative models. Though result quality remains far inferior to e.g. SotA GANs, I consider these important and interesting contributions that likely pave the way for further inquiry into score-based models.

Strengths: Starting from the premise of thinking that score-based generative models are an interesting approach overall, I consider these contributions highly valuable. The authors derive the results through meaningful analysis instead of pulling them out of a hat; furthermore evaluation and validation is very good.

Weaknesses: The only weaknesses I can think of is a lack of practical information on things like how long the models take to train on what hardware, and how long does generating samples take in each resolution/dataset. I would like to see this information in the rebuttal.

Correctness: Yes.

Clarity: The paper is very well written. I like the smooth coexistence of intuition and math.

Relation to Prior Work: Yes.

Reproducibility: Yes

Additional Feedback:

[Author Response · NeurIPS 2020]

We thank all the reviewers for providing valuable feedback in this time of stress. Below we first discuss a new discovery
on evaluation metrics, and then answer specific questions of the reviewers.

**A surprising fact of FID computation.** Because of annealed
Langevin dynamics, the samples contain small Gaussian noise that
is imperceptible to human eyes. After our paper submission, we
discovered that this small Gaussian noise—though hard to detect by
humans—can greatly hurt the FID scores. Therefore, we **denoise**
**the samples by running one step of $\mathbf{x} \leftarrow \mathbf{x} + \sigma_L^2 \mathbf{s}_\theta(\mathbf{x}, \sigma_L)$ and**
**compute the FID scores again.** We provide the new ablation results
in the right figure, with the extra configuration $\text{NCSN}_{1,2,4}$ suggested

by **R3**. We follow the same checkpoint selection method in Table 5 and provide full FID scores below. **The NCSNv2**
**model now obtains much lower FID scores than NCSN**, which aligns better with our visual inspection of samples.
We are surprised by how the FID scores improve for both NCSN and NCSNv2 though **samples before and after this**
**additional denoising step are the same to naked eyes**. We will include these new results in the revision.

|  | NCSN (CIFAR-10) | NCSNv2 (CIFAR-10) ‖ NCSN (CelebA) | NCSNv2 (CelebA) |
|---|---|---|---|
| FID | 27.44 | **10.31** ‖ 17.57 | **9.69** |

**[R1] Is the model memorizing data (like the Eiffel towers in Figure 1)?** In the paper, we
argued from several perspectives that the model is not memorizing data: (i) The test loss and
training loss are comparable to each other (see Figure 12); (ii) Nearest neighbors in the training
dataset do not look the same as samples from the model (see Section C.4.2); and (iii) The model
can generate samples that smoothly interpolate from one to another (see Figure 7 and Section
C.4.3). In the right figure, we additionally provide nearest neighbors (the right column) in $\ell_2$
distance to the two Eiffel towers (the left column) which appeared in Figure 1.

**[R1][R2] Whether EMA has a negative impact on performance?** As **R1** and **R2** noted, EMA stabilizes training but
sometimes may have a slightly worse peak FID score. Because a larger variance gives rise to larger extreme values,
unstable methods naturally lead to a better peak FID score. However, we believe this is an imperfection of the peak FID
metric, rather than an indicator that unstable methods perform better. In fact, as shown in Figure 4 and 11, EMA yields
lower FIDs most of the time and samples with EMA look much more visually appealing than those without EMA.

**[R2] Are all techniques needed for scaling to higher resolution?** From the new ablation results above, we observe
that using all techniques leads to the best performance. We agree that for a specific dataset like CelebA it may not be
necessary to use all 5 techniques to get reasonable results, but one key point of our paper is that using all techniques
make the model **work out of the box** for **a large number of different datasets**, which we demonstrate on many
datasets of different resolutions, including $32^2$, $64^2$, $96^2$, $128^2$ and $256^2$.

**[R2] When does the RefineNet architecture change?** We hope to clarify one confusion: in the ablation study, only
NCSNv2 uses the new architecture and the others use the old one. The new architecture is necessary for using Technique
3 because it assumes an unconditional score network. We can view the impact of this architecture change by comparing
$\text{NCSN}_{1,2,4,5}$ and $\text{NCSN}_{1,2,3,4,5}$ (*i.e.*, NCSNv2) in the ablation results.

**[R2] Writing issues.** Thanks for pointing them out! We will incorporate your suggestions in the revision.

**[R2][R3] Evaluation metrics.** There are many known issues with existing metrics of sample quality, and finding the
right one is still an open problem. We choose FID and $\text{HYPE}_\infty$ as an approximation to the real sample quality. The
discrepancy of FIDs in Figure 5 and Table 5 is because FIDs in Table 5 **are the peak FIDs and are computed on**
**more samples**. The $\text{HYPE}_\infty$ scores are computed on more than 2000 **uncurated** samples, and are better when closer
to 50. "Fakes Error" is the proportion of fake images perceived as real, and "Reals Error" is the opposite.

**[R3] Oversimplified assumptions.** Despite using simplified assumptions, our theory predicts parameters that perform
very well across a large number of complicated real datasets. It is proved by our experiments to be useful and valuable.

**[R3] KDE and Technique 1, 2, 4.** When applied to multi-scale KDE as in the
setting of Figure 2, annealed Langevin dynamics will converge to samples that
**exist in the training dataset**. Training an NCSN makes it possible to generalize
to novel samples. We provide the ablation study of Technique 1, 2, 4 in the above
figure (see $\text{NCSN}_{1,2,4}$), showing that they can improve FID over NCSN even
without EMA (Technique 5).

| Dataset | Device | Sampling time | Training time |
|---|---|---|---|
| CIFAR-10 | 2x V100 | 2 min | 22 h |
| CelebA | 4x V100 | 7 min | 29 h |
| Church | 8x V100 | 17 min | 52 h |
| Bedroom | 8x V100 | 19 min | 52 h |
| Tower | 8x V100 | 19 min | 52 h |
| FFHQ | 8x V100 | 50 min | 41 h |

**[R2][R4] How long the model trains on what hardware, and the sampling speed.** We provide the statistics in the
above table, and will add it to the paper in the next revision. The sampling time is for one mini-batch.

[Meta-Review · NeurIPS 2020]

All reviewers recommend acceptance. Some minor concerns were raised about some missing details and the way the results are presented, but the reviewers agree that the author feedback addresses most of these concerns satisfactorily. R2 points out that the sensitivity of FID to noise has been observed previously in the literature. I recommend that the authors take this into account when they update the manuscript.